# Biswas–Chatterjee–Sen Model Defined on Solomon Networks in (1 ≤ *D* ≤ 6)-Dimensional Lattices

**DOI:** 10.3390/e27030300

**Published:** 2025-03-14

**Authors:** Gessineide Sousa Oliveira, David Santana Alencar, Tayroni Alencar Alves, José Ferreira da Silva Neto, Gladstone Alencar Alves, Antônio Macedo-Filho, Ronan S. Ferreira, Francisco Welington Lima, João Antônio Plascak

**Affiliations:** 1Dietrich Stauffer Computational Physics Lab, Departamento de Física, Universidade Federal do Piauí, Teresina 64049-550, PI, Brazil; sousagessi21@gmail.com (G.S.O.); d.s.m.alencar@gmail.com (D.S.A.); tay@ufpi.edu.br (T.A.A.); ferreiraesol@gmail.com (J.F.d.S.N.); 2Departamento de Física, Universidade Estadual do Piauí, Teresina 64002-150, PI, Brazil; alves.gladstone@gmail.com (G.A.A.); amfilho@gmail.com (A.M.-F.); 3Departamento de Ciências Exatas e Aplicadas, Universidade Federal de Ouro Preto, João Monlevade 35931-008, MG, Brazil; ronan.ferreira@ufop.edu.br; 4Departamento de Física, Centro de Ciências Exatas e da Natureza, CCEN, Universidade Federal da Paraíba, Cidade Universitária, João Pessoa 58051-970, PB, Brazil; 5Departamento de Física, Universidade Federal de Minas Gerais, C. P. 702, Belo Horizonte 30123-970, MG, Brazil; 6Department of Physics and Astronomy, University of Georgia, Athens, GA 30602, USA

**Keywords:** nonequilibrium, phase transition, Monte Carlo simulations, Solomon networks, lower upper critical dimension

## Abstract

The discrete version of the Biswas–Chatterjee–Sen model, defined on *D*-dimensional hypercubic Solomon networks, with 1≤D≤6, has been studied by means of extensive Monte Carlo simulations. Thermodynamic-like variables have been computed as a function of the external noise probability. Finite-size scaling theory, applied to different network sizes, has been utilized in order to characterize the phase transition of the system in the thermodynamic limit. The results show that the model presents a phase transition of the second order for all considered dimensions. Despite the lower critical dimension being zero, this dynamical system seems not to have any upper critical dimension since the critical exponents change with *D* and go away from the expected mean-field values. Although larger networks could not be simulated because the number of sites drastically increases with the dimension *D*, the scaling regime has been achieved when computing the critical exponent ratios and the corresponding critical noise probability.

## 1. Introduction

The critical properties of a system undergoing a second-order phase transition are strongly dependent on the dimension of the underlying lattice. In fact, the lattice dimension is one of the crucial factors to determine the universality class of a particular second-order phase transition (the other factors consist of the dimension of the order parameter and symmetry and range of the particle interactions). Renormalization group theory turned out to be a successful framework to understand the universal aspects of the critical phenomena from the knowledge of the basic microscopic interactions (see, for instance, ref. [1] and references therein).

It is also well established in the literature that each system has its proper lower critical dimension $D_{l}$, below which there is no phase transition, and also an upper critical dimension $D_{u}$, above which the transition is governed by the mean-field exponents [2]. Of course, the upper critical dimension has only a theoretical interest because $D_{u}$ is usually above the three-dimensional space of the physical sample’s realizations [3]. To cite just some examples, we have $D_{l}=1$ and $D_{u}=4$ for the Ising model [1], $D_{l}=2$ and $D_{u}=4$ for the isotropic Heisenberg model [1], $D_{l}=2$ [4] and $D_{u}=6$ [5] for the Ising spun-glass system. More recently, it has been reported that $D_{l}=1$ and $D_{u}=6$ for the majority vote model on regular lattices [6].

The magnetic models cited above, defined on regular Bravais lattices, are well studied in the literature because they are in some sense easier to define and suitable for being simulated using different sorts of computer algorithms and have plenty of physical realizations to be compared to. Nevertheless, magnetic models can also be defined on scaling-free lattices, like networks and specific graphs. In this case, however, it is not possible to assert, a priori, whether the system will or will not have a phase transition. Moreover, when a phase transition is present, one has to carefully investigate whether the transition is of first or second order. Some magnetic systems, like the Ising, Potts and Blume–Capel models, have already been studied on some complex lattices comprising Appolonian, Barabási-Albert, Voronoi-Delauny, and small-world networks [7].

There are also different sorts of dynamical systems, not along the line of the magnetic ones, that are studied on regular lattices. In this case, the determination of the lower and upper dimensions turns out to be a point of theoretical interest. For instance, an interesting result has been obtained for the majority vote model [8] through Monte Carlo simulations in several *D*-dimensional lattices [6]. It has been reported that the model does present an upper critical dimension $D_{u}=6$, despite the majority vote model being in the same universality class as the Ising model for D=2 (recall that $D_{u}=4$ of the Ising class).

In this work, we will address the question of the upper critical dimension of the Biswas–Chatterjee–Sen (BChS) model, in its discrete version, and defined on Solomon networks (SNs). The BChS model is a complex system that takes into account the dynamics of opinions of its individual constituents [9,10,11]. Intrinsic to the system is the fact that the individuals in a community can not only influence their neighbors but also be influenced by them. Accordingly, the individual opinion variables evolve according to pair interactions that are allowed to be positive or negative. The interaction signs are then modeled by a single noise probability *q* that represents the fraction of negative interactions. This noise probability plays a crucial role in the dynamics of this system. In fact, *q* is the temperature equivalent in magnetic models.

On regular lattices, and also in some special networks, the BChS model has a second-order phase transition at a critical value $q_{c}$, with critical exponents different from the usual magnetic models and dependent on the particular chosen lattice [12]. For a recent review on the progress already made in this model, see Ref. [13].

With respect to scaling free lattices, in the direction of treating opinion dynamics, a more realistic environment has been proposed by Sorin Solomon [14,15], who considers two different lattices. One lattice reflects one kind of environment, for instance, the home place, and the other lattice a different situation, for instance, the workplace. In practice, labeling with *i* the sites in the workplace lattice, a random permutation P(i) of the order already established in this lattice provides the sites in the home place lattice. The actuality in this process resides in the fact that the neighbors at home differ from the neighbors at the workplace. Thus, like in the biblical story of King Solomon, in such a construction, each individual is equally shared by two lattices. So, the net interaction of the relevant variables defined at site *i* turns out to be a sum of the corresponding interactions of the site *i* with its neighboring sites on the workplace lattice, plus the interactions with the neighbor sites of P(i) on the home place lattice. It should be stressed that the increase in the connectivity of each site *i* makes the SNs close to small-world networks [16,17].

The BChS model has been previously studied in D=1, D=2, and D=3 SNs [12,18]. The critical exponents of the second-order phase transition depend on the dimension of the lattices *D*, as expected from general renormalization group arguments. Since the majority vote model presents an upper critical dimension, it would be worthwhile to see whether the BChS model in higher dimension SNs will have critical exponents in the direction of the mean-field ones.

Thus, in the present work, the BChS dynamical system on SNs has been studied through extensive Monte Carlo simulations by considering lattices in D=4–6 dimensions. In the Section 2, we have the definition of the BChS model in discrete opinion dynamics; the themodynamic-like variables that are used in describing the phase transition; and the corresponding finite-size-scaling relations, together with some details of the Monte Carlo simulations. The results are discussed in Section 3, and some concluding remarks are summarized in the Section 4.

## 2. Model, Thermodynamic-like Variables, and Simulations

The discrete version of the BChS model has already been extensively described in previous papers. However, for completeness, we will give a detailed description below of the model and its relevant variables, as well as the thermodynamic-like quantities from which the phase transition properties can be obtained. Details of the Monte Carlo simulations on lattices of linear size *L* and the finite-size-scaling relations of the thermodynamic-like quantities used are also discussed to treat the model in several dimensions, namely D=4, 5 and 6.

### 2.1. Biswas–Chatterjee–Sen Model

The BChS dynamical system, defined in different regular lattices and also different networks, has already been studied using Monte Carlo simulations (see, for instance, Refs. [9,10,12,18,19]).

To start with, consider first an SN with $N=L^{D}$ sites. A sketch of a SN network in D=2 is reproduced in Figure 1. At each site *i* of the network, we have an agent, or individual, carrying an opinion variable. This opinion variable changes with time, and at time *t*, the opinion has a definite value $o_{i}\left(t\right)$. In the discrete version adopted herein, these opinion variables of all the agents may assume only one of three different values, which can be set as −1, 0, or +1. Each pair of agents *i* and *j*, which can be a nearest neighbor or even distant neighbors, has a bond pair that is initially ascribed a positive affinity $\mu_{ij}=+1$. However, during the dynamical process of updating the opinion variables, this affinity can be turned negative, $\mu_{ij}=-1$, with a probability *q*. In fact, this probability *q* acts as an external noise, modeling the local discordances. For instance, with a negative affinity, two agents with the same sign of opinion values, could end, after the updating, with different signs of opinion values.

Thus, the sum of the site *i* opinion variable and the affinity times the opinion variable of its neighbor *j* is the updated opinion variable. The same is carried out for the neighbor *j*. We then have(1)$$\begin{array}{ccc} o_{i}+\mu_{ij}o_{j} & \to & o_{i}\;, \end{array}$$(2)$$\begin{array}{ccc} o_{j}+\mu_{ij}o_{i} & \to & o_{j}\;. \end{array}$$
$o_{i}$ and $o_{j}$ in the right hand side of the above equations are the updated opinion states of the pair ij. Table 1 conveys a summary of the above symbols adopted in the model and their respective description.

The dynamical rules of the BChS model to update the opinion variables $o_{i}\left(t\right)$ are as follows.

•
**Initialization of the system**


(i)At time t=0, an initial configuration is constructed by randomly assigning one of the three opinion states, namely +1, 0 or −1, to each site *i* of the workplace lattice. As a result, we have $\left\{o_{i}\left(t=0\right)\right\}=\left\{o_{i}\right\}$. A particular value of the noise probability *q* is chosen.


**Beginning of the Monte Carlo step execution**


(ii)A permutation P(i) is carried out in the workplace lattice in order to generate the home place lattice. However, instead of creating a real new lattice, in this procedure, it simply stores, for each site *i*, a list of its neighbors in the workplace and also in the home place. As a result, for each agent, one has a total of 2*D nearest neighbors*.


**Selection of the nearest neighbor and updating of the pair of opinion variables**


(iii)For a given site *i*, one of its 2D nearest neighbors *j* is randomly selected (note that the selected neighbor may be in the workplace or home place lattice); an affinity $\mu_{ij}=+1$ is ascribed for this pair bond; and a random number *r* is sorted. If r<q, the affinity is changed to $\mu_{ij}=-1$.(iv)New values of the opinion variable of both the nearest neighbor sites $o_{i}$ and $o_{j}$ are then assigned according to Equations (Equation 1) and (Equation 2).


**Renormalizing opinion variables**


(v)It may be that the opinion variables are out of the integer range [−1,+1]. When this happens, they are automatically returned to the desirable interval by rescaling as $o_{i}=+1$ if $o_{i}>+1$, or $o_{i}=-1$ if $o_{i}<-1$.


**End of the Monte Carlo step execution**


(vi)Every site *i* of the workplace lattice is sequentially swept and updated according to the rules above. Thus, one whole sweep of the network constitutes one Monte Carlo step per site (MCS). This is, by definition, the unit of the Monte Carlo time, meaning that the new network configuration corresponds to time t+1, while the previous one was at time *t*.

### 2.2. Thermodynamic-like Variables

The convenient order parameter *O* that has been usually defined for this model consists of averaging the opinion variables $o_{i}\left(t\right)$ over all individuals, i.e.,(3)$$O=\left|\sum_{i=1}^{N}o_{i}\left(t\right)\right|/N,$$
where *N* is the total number of sites of the SN. In the above equation, the time *t* is chosen to be large enough for the system to have reached its stationary state. In contrast to ordinary magnetic systems, where one has a thermal-driven phase transition, in the present model, one has a kind of a random-configuration-driven phase transition. The phase transition here means that for $q<q_{c}$, O≠0 and an ordered phase is established, while for $q>q_{c}$, O=0 and the system is in a disordered phase. At exactly $q=q_{c}$, a second-order phase transition takes place.

From the order parameter defined in (Equation 3), it is possible to construct its magnetic-like variables, such as the order parameter fluctuation $O_{f}\left(q\right)$ or susceptibility (being analogous to the magnetic susceptibility), its reduced fourth-order Binder cumulant $O_{4}\left(q\right)$, the derivative of the cumulant with respect to the noise probability *q*, and so on. These observables, from which it is possible to fully categorize the possible phase transition, can be formally expressed as(4)$$\begin{array}{ccc} O(q) & = & [{\langle O\rangle }_{t}{]}_{av}, \end{array}$$(5)$$\begin{array}{ccc} O_{f}\left(q\right) & = & N[{\langle O^{2}\rangle }_{t}-{\langle O\rangle }_{t}^{2}{]}_{av}, \end{array}$$(6)$$\begin{array}{ccc} O_{4}\left(q\right) & = & 1-[\frac{{\langle O^{4}\rangle }_{t}}{3{\langle O^{2}\rangle }_{t}^{2}}{]}_{av}, \end{array}$$
where ${\langle \cdots \rangle }_{t}$ stands for different time averages (which are computed after the system has reached the stationary state), and ${\left[\cdots \right]}_{av}$ means the averages taken over different initial configurations.

### 2.3. Finite-Size Scaling Relations and Monte Carlo Simulations

All the variables above can be obtained as a function of *q*, for several SN network sizes *L*, in a particular spatial dimension *D*. The criticality or any possible phase transition exhibited by the model can thus be inferred by analyzing the behavior of Equations (Equation 4)–(Equation 6) close to the transition point $q_{c}$ as a function of *L*. This analysis is carried out by using the finite-size scaling hypothesis. This hypothesis states that, for large system sizes *L*, the relevant variables above should follow a power-law behavior of the form (for further details see, for instance, Refs. [20,21,22])(7)$$\begin{array}{ccc} O(q) & = & L^{-\beta /\nu }f_{O}\left(x\right), \end{array}$$(8)$$\begin{array}{ccc} O_{f}\left(q\right) & = & L^{\gamma /\nu }f_{O_{f}}\left(x\right), \end{array}$$(9)$$\begin{array}{ccc} O_{4}\left(q\right) & = & f_{O_{4}}\left(x\right), \end{array}$$(10)$$\begin{array}{ccc} \frac{dO_{4}}{dq} & = & L^{1/\nu }f_{d}\left(x\right), \end{array}$$(11)$$\begin{array}{ccc} q_{c}\left(L\right) & = & q_{c}+q_{0}L^{-1/\nu }, \end{array}$$
where β, ν, and γ are the critical exponents of the order parameter, correlation length, and fluctuation of the order parameter, respectively. $f_{k}\left(x\right)$ are the respective scaling functions, with the scaling variable $x=L^{1/\nu }\left(q-q_{c}\right)$ and $k=\{O,\;O_{f},\;O_{4},\;d\}$. $\frac{dO_{4}}{dq}$ is the derivative of the fourth-order Binder cumulant with respect to *q*. In Equation (Equation 11), the critical noise probability $q_{c}\left(L\right)$, for a given lattice size *L*, can be chosen, for example, the estimate of the position of the peak of the fluctuation $O_{f}$ or the position of the maximum value of the magnitude of the derivative $\frac{dO_{4}}{dq}$. In this equation, $q_{c}$ is the critical noise probability in the thermodynamic limit with $q_{0}$ as a non-universal constant. When the large lattice regime has not been reached, corrections to the finite-size scaling can be implemented in the above equations with additional exponents and fitting parameters.

The regular procedure to determine the transition point and the corresponding exponents are now as follows. A ln-ln plot of the maximum value of the magnitude of the derivative $\frac{dO_{4}}{dq}$, as a function of *L*, allows one to extract the critical exponent 1/ν. From the position $q_{c}\left(L\right)$ of these maxima, as well as from the position of the peaks of the order parameter fluctuation $O_{f}$, and using Equation (Equation 11) with 1/ν in hand, one obtains the critical noise probability $q_{c}$ in the thermodynamic limit. Back to Equations (Equation 7) and (Equation 8), ln-ln plots of $O(q_{c})$ and $O_{f}\left(q_{c}\right)$ furnish the exponent rations β/ν and γ/ν, respectively. The peak values $O_{f}\left(q_{max}\right)$ of the fluctuation $O_{f}$ gives also an additional estimate of γ/ν.

The Monte Carlo simulations were performed according to the algorithm described above in items (i)–(viii) on several SNs of finite sizes *L*. Since the number of sites exponentially increases with the dimension, smaller sizes were simulated as *D* increased. In this way, for D=4, we have L=4, 5, 6, 7, 8, 9 and 10; for D=5 we have L=3, 4, 5, and 6; and for D=6 we have L=2, 3, 4, 5, and 6. All relevant quantities in Equations (Equation 4)–(Equation 6) were computed as a function of the noise probability *q*, and, close to the transition region, the noise probability steps were chosen as Δq=0.001.

In all simulations, in order to compute the desired observables, several MCS were performed on the SNs. In general, 120 different SN realizations for each network size were simulated to make the quench averages. For each network replica, $10^{5}$ MCS were carried out to let the system evolve to a stationary state, and then another $10^{5}$ MCS were carried out to collect the $10^{5}$ values of the opinion variables used to measure the desired observables. Error bars were estimated by using the jackknife resampling technique [21,23].

## 3. Results and Discussion

As a matter of example, the general behavior of the relevant quantities for this model in D=4, namely *O*, $O_{f}$ and $O_{4}$, given by Equations (Equation 4)–(Equation 6), are, respectively, displayed in the three panels (a), (b) and (c) of Figure 2. The different lattice sizes are listed in the legend of Figure 2a. In all these panels, only the lines of the data are shown for a clearer visualization of the dependence of the respective quantities as a function of the noise probability *q*. From the behavior of *O*, $O_{f}$ and $O_{4}$ close to the transition region, it is clear that the system undergoes a phase transition, since the derivatives of *O* and $O_{4}$ with respect to *q* increase in magnitude as the lattice size increases, as does the peak of $O_{f}$ (note that, for a better view of the peaks, the vertical axis of Figure 2b gives $O_{f}$ in a logarithm scale). In addition, the fourth-order Binder cumulants $O_{4}$ cross at the same region, which is equivalent to the critical noise transition. Similar patterns, as highlighted in this figure, have been previously obtained for D≤3 and are also obtained for D=5 and 6. However, as will be seen below, the cumulant crossings for higher dimensions D≥5 are not as well defined as for the case D≤4. Some details about extracting the critical properties of the model from this kind of data are discussed below.

Let us first consider the fourth-order Binder cumulant of the order parameter $O_{4}$, as a function of the disorder parameter *q*, which is depicted in the left panels of Figure 3 for several lattice sizes *L* and dimensions *D*. In panels Figure 3a,c,e, we have D=4, D=5, and D=6, respectively. One can clearly see from panel Figure 3a for D=4 that the system undergoes a second-order phase transition, since the cumulants tend to cross at the same value *q*, which corresponds to the critical disorder parameter $q_{c}$ [20]. In the axis scales used in Figure 3a, one can make a rough estimate of the critical noise and the universal value of the Binder cumulant $q_{c}=0.204\left(5\right)$ and $O_{4}^{*}=0.291\left(2\right)$, respectively. It can also be noticed from Figure 3c,e (D=5, D=6) that we do not have a clear estimate either of $q_{c}$ or $O_{4}^{*}$. This is a result of the smaller lattices that were used, because the number of sites to be simulated exponentially increases with *D*. Nevertheless, from the inset of the left panels, it is seen that the slope of $O_{4}$ systematically increases with the lattice size. From the maximum value of the slope of $O_{4}$ with respect to *q*, one is able to obtain the correlation length exponent 1/ν. The magnitude of the derivative $dO_{4}/dq$ was numerically computed from the original simulated data. The ln-ln plot of this derivative, as a function of the lattice size *L*, is illustrated in the main graphs of the right panels of Figure 3 for each considered dimension *D*. The slope of the linear fit of the data, according to Equation (Equation 10), corresponds to the exponent ratio 1/ν. The values of 1/ν are given in the proper figures. In dimensions D=5 and D=6, the smaller lattices have not been taken in the linear fit.

The position $q_{c}\left(L\right)$ of the peak of the derivative $dO_{4}/dq$ can also be used to estimate the critical value $q_{c}$ in the thermodynamic limit. The behavior of $q_{c}\left(L\right)$ as a function of $L^{-1/\nu }$ is shown in the inset of Figure 3b,d,f. In these insets, the two smaller lattices in dimension D=4 and the smallest lattice in dimensions D=5 and D=6 have been omitted in order to have only the data with better alignment. The results using a linear fit are given in the respective legends. It is worthwhile to stress that, when considering all available lattices, the data are not quite aligned, mainly for higher dimensions. In this case, corrections to scaling could be used, with power law behavior (and considering an extra parameter as the correction-to-scaling exponent) and logarithmic corrections as well. In all cases, however, the extrapolated value of $q_{c}$ are, within the error bars, equivalent to the linear fits shown in Figure 3b,d,f.

Although for D=5 and D=6, the value of the extrapolated $q_{c}$ is not clearly seen in the cumulants crossings, for D=4, the extrapolated $q_{c}=0.203\left(1\right)$ value is, within the error bars, comparable to crossings in Figure 3a, namely $q_{c}=0.204\left(1\right)$. However, good estimates of the universal value $O_{4}^{*}$ are still quite imprecise from Figure 3c,e. This means that for the lattice sizes used in the present model, finite size effects turn out to actually be important to obtain $O_{4}^{*}$ in dimensions D>4.

The estimate of the critical noise probability $q_{c}$ allows us now, through Equations (Equation 7) and (Equation 8), to evaluate the critical exponent ratio β/ν and γ/ν by just computing the order parameter $O(q_{c})$ and its fluctuation $O_{f}\left(q_{c}\right)$ at $q_{c}$ for different lattice sizes *L* and spatial dimensions *D*. In the case of $O_{f}$, we also have its maximum value $O_{f}\left(q_{max}\right)$ at the noise probability $q_{max}$. Figure 4 depicts the ln-ln plot of these quantities as a function of the system size *L*. The corresponding slopes of the linear fit give the respective critical exponent ratios and are written in the legends of Figure 4a–c for D=4, D=5, and D=6, respectively. Although $O_{f}\left(q_{c}\right)$ and $O_{f}\left(q_{max}\right)$ have different values for each lattice size, the γ/ν exponent estimates agree well within the error bars.

The noise probability $q_{max}$, at which the fluctuation variable $O_{f}$ exhibits a maximum, can be interpreted as another $q_{c}\left(L\right)$. Accordingly, new estimates of the critical value $q_{c}$ can be performed from additional linear fits. Figure 5 displays the values of $q_{c}\left(L\right)$ obtained accordingly as a function of $L^{-1/\nu }$ for the different lattice dimension *D*. The full lines in Figure 5 give the best linear fit with the results of $q_{c}$, for each dimension, as displayed in the legend. The error in $q_{max}$ has been adopted as 0.001, the interval used to increase the noise probability in the simulations.

Finally, Figure 6 also shows, in D=4 dimensions, the data collapse obtained for the rescaled order parameter $OL^{\beta /\nu }$ in panel (a) for the rescaled fluctuation of the order parameter $O_{f}L^{-\gamma /\nu }$ in panel (b), and for the rescaled reduced Binder cumulant $O_{4}$ in panel (c). All quantities are given as a function of the rescaled probability displacement $\left(q-q_{c}\right)L^{1/\nu }$. As in the panels of Figure 2, only the lines of the actual Monte Carlo data have been plotted for a better evaluation of the collapse and the accuracy of the critical quantities. The corresponding lattice sizes *L* are listed in the legend of Figure 6a, where the two smaller lattice sizes have been omitted in the data collapse. Apart from the order parameter for small values of the noise probability, the excellent agreement with the scaling relations given in Equations (Equation 7), (Equation 8), and (Equation 11) is apparent. This is a clear indication that the evaluation of the critical exponents ratio β/ν, γ/ν, and 1/ν are reasonably accurate.

Despite the finite-size effects being more pronounced as the lattice dimension increases, similar results to those illustrated in Figure 2 and Figure 6 are also obtained for higher dimensions *D* using the corresponding critical values of the noise probability and exponent ratios.

## 4. Concluding Remarks

The discrete version of the Biswas–Chatterjee–Sen model, defined on 4-, 5- and 6-dimensional Solomon networks, has been studied through extensive Monte Carlo simulations as a function of the local consensus controlling probability. From the computer simulation data and from the scaling behavior of the thermodynamical-like quantities used, it has been seen that the model really undergoes a well-defined second-order phase transition for all treated dimensions.

Table 2 summarizes the numerical results of those criticality observed for the present values of lattice dimension *D*, namely D=4, 5, and 6, together with the results previously obtained for D=1, 2, and 3 from Refs. [12,18]. For $q_{c}$, we have taken the average from the values conveyed in Figure 3 and Figure 5. An inspection of this table readily shows the following.

(1)The critical noise probability, given the error bars, does not systematically change over the present studied dimensions. This is indeed an intriguing and unexpected result, since in spin models, the number of nearest neighbors effectively increases the transition temperature. It should also be emphasized that the BChS model on regular lattices has a critical noise probability that increases with the lattice dimension [10]. It is also improbable that, due to the limitations of the system sizes considered herein, by increasing the dimension of the networks, different sets of neighbors of an agent start to overlap. In fact, overlapping neighborhoods are not allowed by the computer code. It seems that in SNs, the spatial dimension is not an important issue regarding the location of the critical point transition.(2)The critical exponents, on the other hand, do depend on the lattice dimension, as is expected from universality arguments. In this case, the exponent ratios systematically increase as *D* increases.(3)The hyperscaling relation D=2β/ν+γ/ν is, within the error bars, not violated and is actually satisfied in all of the studied dimensions. The same holds for the equivalent relation Dν=2 using the correlation length exponent, although in the latter case, one observes a larger error.(4)The lower critical dimension for this model is 0, but from the present simulations, there is no indication of an upper critical dimension from where the mean-field exponents are valid. We can note from Table 2 that the closest mean-field exponents one obtains are for the expected dimension, D=4.

The crossings of the cumulants, which in general give further estimates of $q_{c}\left(L\right)$, could not be considered in this dynamical system for D≥4, although for D=3, it was possible to obtain good estimates of $q_{c}$ and $O_{4}^{*}$ from them [12] (also using logarithm-like corrections). In the present dimensions, even and odd lattice sizes follow different uniform trends to reach the thermodynamic limit value. However, since larger lattices could not be simulated, estimates for the critical quantities turned out to be very poor due to few data being available for the proper fits.

As a final remark, it should be stressed that the lattice sizes utilized in this study could perhaps not be as large as one should desire for the proper finite-size scaling to be valid. Of course, still larger lattices should be used in order to obtain more reliable estimates. However, due to the quality of the fits with the present system sizes, we believe that quantitative changes in the values of the exponents and their ratios cannot be so sensitive to turn the exponents closer to the MF values for D>4. 

## Figures and Tables

**Figure 1 entropy-27-00300-f001:**
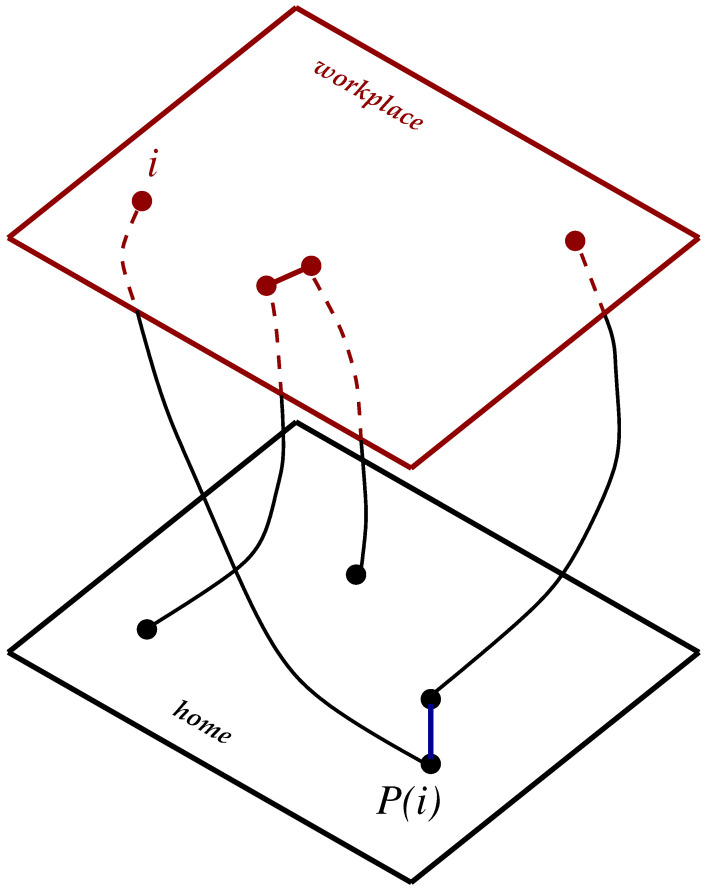
(Color online) Sketch of a two-dimensional Solomon network. The upper lattice is the workplace, and the bottom lattice is the home place. Circles represent agents, and linked circles are nearest neighbors. P(i) is the permutation place in the home lattice of the *i* site in the workplace lattice. Note that nearest neighbors in one lattice are not necessarily nearest neighbors in the other lattice.

**Figure 2 entropy-27-00300-f002:**
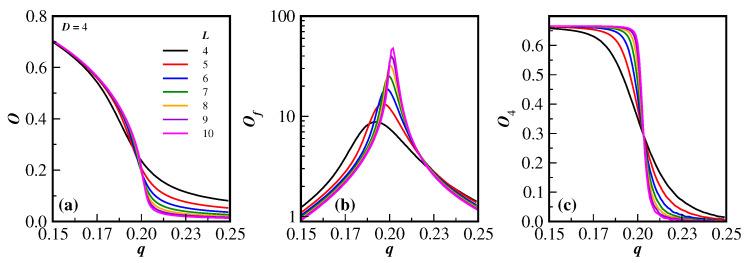
(Color online) The order parameter *O*, the fluctuation of the order parameter $O_{f}$, and the fourth-order Binder cumulant $O_{4}$ are, respectively, given in (**a**–**c**) as a function of the noise probability *q* for the model on D=4 dimensions and lattice sizes in the range 4≤L≤10. The legend in the left panel also applies to the two right panels.

**Figure 3 entropy-27-00300-f003:**
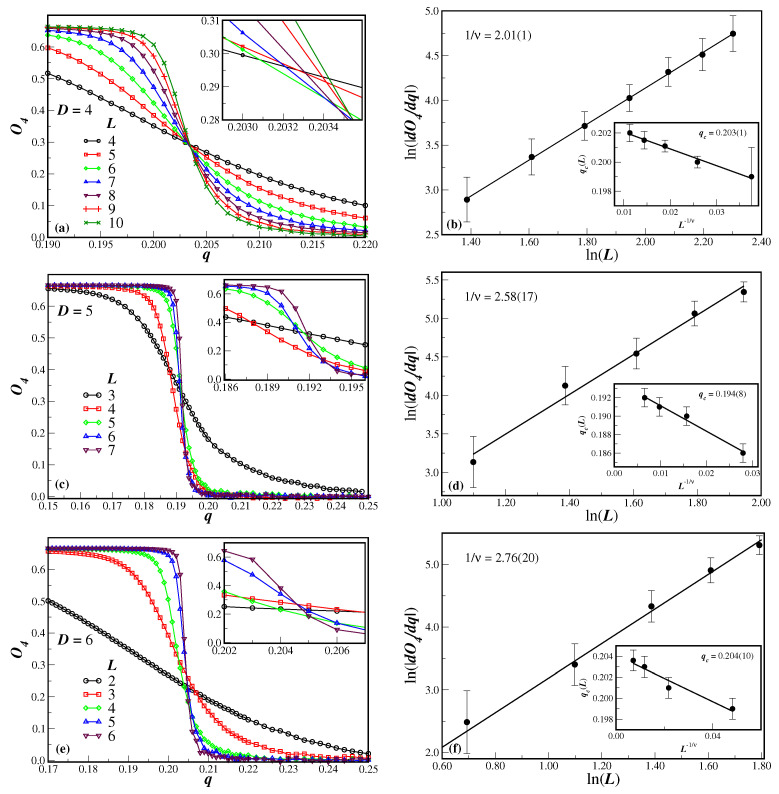
(Color online) The fourth-order Binder cumulant of the order parameter $O_{4}$, as a function of the disorder parameter *q*, for several finite lattices of size *L* is shown in (**a**) for D=4, in (**c**) for D=5, and in (**e**) for D=6 (the corresponding insets are an amplified view of the cumulant crossings close to the critical noise probability). The lines in the left panels are just a guide for the eyes. The main graphs in the right panels (**b**,**d**,**f**) are the ln-ln plots of the cumulant derivative with respect to *q* as a function of the lattice size *L* for each dimension *D*. In these cases, the full lines are linear fits to the data to obtain the exponent 1/ν. The inset in the right panels illustrates the behavior of $q_{c}\left(L\right)$ as function of $L^{-1/\nu }$, with ν coming from the previous fits, and the full lines are also linear fits to obtain $q_{c}$.

**Figure 4 entropy-27-00300-f004:**
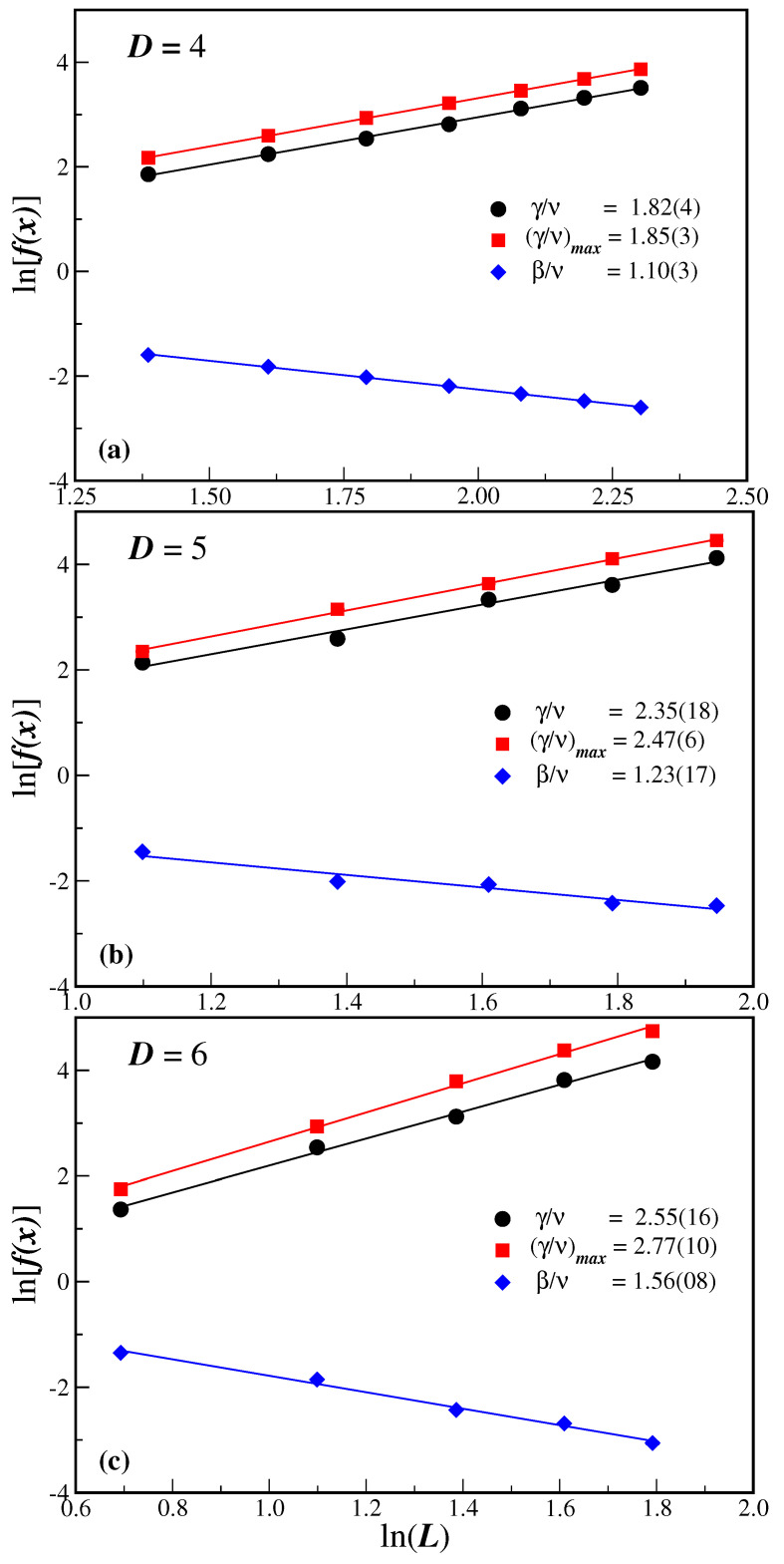
(Color on-line). Ln-ln plots of f(x) as a function of lattice size *L* for D=4 in (**a**), D=5 in (**b**), and D=6 in (**c**). The function f(x) is, from top to bottom in each panel. the susceptibility of the average opinion $O_{f}$ at the estimated $q_{c}$, $f\left(x\right)=O_{f}\left(q_{c}\right)$; the maximum value of the susceptibility at $q_{max}$, $f\left(x\right)=O_{f}\left(q_{max}\right)$; and the average opinion at the estimated critical disorder, $f\left(x\right)=O\left(q_{c}\right)$. The full lines are the best linear fits according to Equations (Equation 7) and (Equation 8), with the corresponding slopes being the critical exponent ratios β/ν and γ/ν, respectively. The legends convey the obtained values of the exponent ratios for each dimension *D*. In all cases, the error bar estimates are smaller than the symbol sizes.

**Figure 5 entropy-27-00300-f005:**
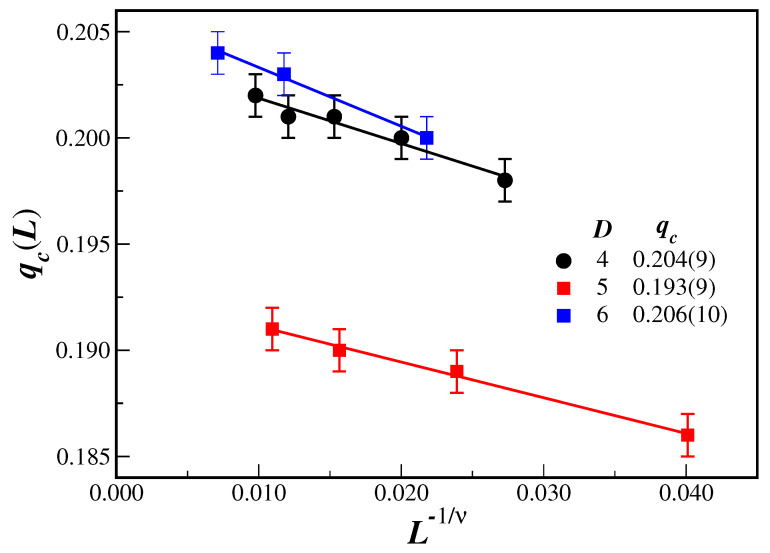
(Color online) Position $q_{c}\left(L\right)=q_{max}$ of the maximum value of the fluctuation of the order parameter $O_{f}$ as a function of lattice size $L^{-1/\nu }$ for different dimensions *D* (given in the legend). The full lines are linear fits with the extrapolated values of $q_{c}$ also given in the legend. Only the larger lattices have been considered.

**Figure 6 entropy-27-00300-f006:**
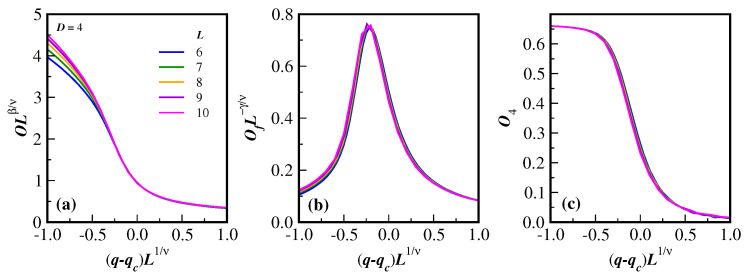
(Color online) The corresponding data collapse of the quantities displayed in Figure 2 using the finite-size scaling relations (Equation 7), (Equation 8), and (Equation 11), which, combined with numerical data in Table 2, are given in panels (**a**–**c**). The excellent data collapse in figures (**a**–**c**) shows that the exponents calculated via finite-size scaling relations are correct.). The legends in the left panels also apply to the two right panels.

**Table 1 entropy-27-00300-t001:** Symbols used in the definition of the BChS model on SNs and their respective descriptions.

Variable	Description
*i*	workplace site of a Solomon network
$o_{i}\left(t\right)$	opinion variable of the agent at site *i* at time *t*
$\mu_{ij}$	affinity of the bond pair ij in either lattice
*q*	noise probability of turning the affinity negative
$\left\{o_{i}\right\}$	represents the configuration of the all opinion variables at time *t*

**Table 2 entropy-27-00300-t002:** Critical noise probability $q_{c}$ and respective critical exponent ratios 1/ν, β/ν, and γ/ν of the BChS model on SNs of different dimensions *D*. The results obtained for D=1, 2, and 3 on the same networks come from Refs. [12,18]. The exponent ratio ${\left(\gamma /\nu \right)}_{max}$ is the result from the maximum value of the order parameter susceptibility. The last two columns give the hyperscaling relations. The last row gives the mean-field (MF) results. Error bars are statistical only.

*D*	$q_{c}$	1/ν	β/ν	γ/ν	${\left(\gamma /\nu \right)}_{\max}$	D=2β/ν+γ/ν	Dν
1	0.215(2)	0.52(5)	0.238(7)	0.511(5)	0.524(6)	1.00(2)	1.92(10)
2	0.216(2)	0.92(4)	0.53(4)	1.02(4)	1.06(7)	2.1(1)	2.17(5)
3	0.1950(7)	1.63(10)	0.789(3)	1.312(8)	1.372(50)	2.95(6)	1.84(15)
4	0.204(5)	2.01(1)	1.10(13)	1.82(04)	1.85(03)	4.00(7)	1.99(8)
5	0.194(8)	2.58(17)	1.23(17)	2.35(18)	2.47(06)	4.80(21)	1.94(9)
6	0.205(10)	2.76(20)	1.56(08)	2.55(16)	2.77(10)	5.67(44)	2.17(10)
MF	—	2	1	2	2	4	2

## Data Availability

Data is contained within the article.

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
