# Peer review of "Biswas–Chatterjee–Sen Model Defined on Solomon Networks in (1 ≤ D ≤ 6)-Dimensional Lattices"

_entropy, 2025, doi:10.3390/e27030300_

Round 1

Reviewer 1 Report

Comments and Suggestions for Authors

The authors study the kinetic exchange opinion model (BChS) in Solomon network in higher dimensions. They report a peculiarity in the upper critical dimension, particularly the absence of it, which is surprising when compared to the other such models of opinion formation.

The findings they report are certainly interesting. However, I have some points that the authors should address first:

  1. I don't quite follow why the critical noise value would be a non-monotonic function of the dimension. When we study such spin-like models in higher dimensions, usually the critical point magnitude goes up. Indeed, it is reflected even in the mean field calculations that the critical point is a monotonic function of the coordination number. I also note that given the error bars in the critical noise value, it is even hard to tell if that changes at all over these dimensions. That is also rather surprising. The authors should comment on that.
  2. I am also not quite clear on the algorithm for simulation of the model in the Solomon network. First of all, they don't quite use two lattices, but choose two sets of neighbors. Is it established that the two things are equivalent in all cases? Even so, how do they tackle overlapping neighborhoods? I guess, particularly in the higher dimensions, such overlaps are possible. Do they explicitly exclude such choices?
  3. They report some logarithmic corrections to scaling. The data presented here is not quite sufficient to reach such a conclusion. They should try to increase the system size and ensemble averaging to confirm such an unusual observation in this model, or mention that such conclusions are not completely clear from the present data.

Overall, these points need to be addressed satisfactorily for further consideration.

Author Response

Comments and Suggestions for Authors
The authors study the kinetic exchange opinion model (BChS) in Solomon network in
higher dimensions. They report a peculiarity in the upper critical dimension, particularly
the absence of it, which is surprising when compared to the other such models of opinion
formation.
The findings they report are certainly interesting. However, I have some points that the
authors should address first:
We thank the reviewer for a positive evaluation of our paper.
1- I don’t quite follow why the critical noise value would be a non-monotonic function
of the dimension. When we study such spin-like models in higher dimensions, usually the
critical point magnitude goes up. Indeed, it is reflected even in the mean field calculations
that the critical point is a monotonic function of the coordination number. I also note that
given the error bars in the critical noise value, it is even hard to tell if that changes at all
over these dimensions. That is also rather surprising. The authors should comment on that.
This is indeed an intriguing and unexpected result for this model. Since our previous
results on D = 2, 3 we have noticed this fact. Unfortunately, we do not have yet a clear
explanation for this strange result. Mainly because in regular lattices q c do increase with D.
So, it is not a question of the model, but a property of the network itself. We also do not
believe that overlapping neighbors (item 2 below and Revisor 2 comment) could be a cause
of this behavior, because the computer code does not allow such overlapping.
We have thus commented this question in the new item 1 of the last section of the revised
paper
2- I am also not quite clear on the algorithm for simulation of the model in the Solomon
network. First of all, they don’t quite use two lattices, but choose two sets of neighbors. Is
it established that the two things are equivalent in all cases?
Yes, the two methods are equivalent, as has been shown in Ref.[11] for lower dimensions (we
expect it to be valid for higher values of D).
Even so, how do they tackle overlapping neighborhoods? I guess, particularly in the higher
dimensions, such overlaps are possible. Do they explicitly exclude such choices? Yes, such
choices of overlapping are avoided in the program code.
2We have made some additional explanations on this point in the end of subsection 2.1
taking also into account similar suggestions from reviewer 2 below.
3- They report some logarithmic corrections to scaling. The data presented here is not
quite sufficient to reach such a conclusion. They should try to increase the system size and
ensemble averaging to confirm such an unusual observation in this model, or mention that
such conclusions are not completely clear from the present data. Overall, these points need
to be addressed satisfactorily for further consideration.
The referee is fully right concerning the logarithmic corrections. In fact, larger lattices
have to be considered to reach such conclusion. We have thus revisited the data and,
neglecting the smallest lattice for D = 5 and D = 6, and the two smaller lattices for D = 4,
a linear fit was able to provide a reasonable value of q c . Despite of this new procedure, there
has been no significant change in the extrapolated value of q c when considering all available
lattice sizes with the use of logarithmic corrections and even power law corrections.
Figures 2(b), (d) and (f), and Figure 4, have been changed accordingly and the new data so
obtained have been included in Table I. Corresponding changes have been also implemented
throughout the text namely: the abstract; 3rd paragraph of section 3; the end of the 4th
paragraph in section 3; the 3rd paragraph in the last section of the previous manuscript has
been deleted; and the 6th paragraph of section 3.

Reviewer 2 Report

Comments and Suggestions for Authors

Author Response

I.
REPORT REVISOR 2
The authors report here the results of extensive Monte Carlo simulations in the discrete
version of the Biswas-Chatterjee-Sen model, defined on 4-, 5- and 6-dimensional Solomon
networks, and their (finite size) transition behavior as a function of the local consensus
controlling probability. The results (including their earlier results for Dimensions 1-3) are
comprehensively summarized in the consolidated table 1. The results are surely intriguing.
Increase in the number of nearest-neighbors (increasing dimension of the hypercubic Solomon
Networks/lattices) effectively increases the transition temperature, and the critical exponent
values depend on the lattice dimension (as is expected from universality arguments). The
results are surely publishable, after some added clarifications on the points mentioned next.
1The unusual observations reported by them are that the exponent seem to go away from
mean field values with increasing D (dimension). Also one notes from the table that they
find a non-monotonicity of the critical point q c with increasing D; considering the error bars,
its highest value had been perhaps at D = 1 and became minimum perhaps at D = 3 (then
increased with higher values of D). Is it possible that, due to limitations of system sizes
considered, with increasing dimension different sets of neighbors of an agent in two different
copies of the Solomon network start overlapping? Some clarifications on this point may be
added.
We thank the reviewer for a positive evaluation of our paper.
It is also interesting the proposed question of overlapping neighbors being the cause of
obtaining the same critical noise parameter for different dimensions. However, as it is
stressed in the new version of the manuscript (particularly item 1 in the last section and
just after describing the algorithm in the end of section 2.1) the computer code does not
allow such overlapping of the sites. It seems indeed a question of the SNs, because the
BChS model on regular lattices do increase q c with D. Some clarifications concerning these
comments are given in item 1 of last section and in the end of subsection 2.1.

Reviewer 3 Report

Comments and Suggestions for Authors

Author Response

I.
REPORT REVISOR 3
The manuscript investigates the Biswas-Chatterjee-Sen (BCS) model on Solomon net-
works across different dimensions (1 ≤ D ≤ 6). The study employs Monte Carlo simula-
tions to model the stochastic dynamics of opinion formation in Solomon networks. Finite-size
scaling analysis is used to determine the critical noise probability and to extract the critical
exponents. Comparisons are made with known results from lower-dimensional cases and
classical models like the Ising model.
The main findings indicate that the BCS model undergoes a well-defined second-order
phase transition for all studied dimensions (D = 1 to 6). The critical noise probability does
not strongly depend on dimensionality, contrasting with traditional spin systems where an
increase in nearest-neighbor interactions lowers the transition temperature. Additionally,
critical exponents systematically increase with increasing dimension (D), and the hyperscal-
ing relation D = 2γ/ν + 1D is verified within the margin of error in all studied dimensions.
The manuscript is clear and well written, provides a thorough investigation into the crit-
ical behavior of the BCS model on Solomon networks, confirming the presence of a second-
order phase transition. These findings contribute to the understanding of phase transitions
in complex networks, particularly in systems governed by stochastic opinion dynamics. This
research extends knowledge on critical phenomena in non-equilibrium systems, highlighting
the effects of network topology on phase transitions. the analysis is conducted in a reason-
able way and the discussions are expected to be beneficial for the readers. Based on my
impressions and comments about the work, I recommend the article for publications. I just
have a few suggestions to clarify some figures to improve the manuscript:
1. Structuring for Readability
Use subsections or bullet points to differentiate key steps clearly.
Group related steps under thematic headings, such as:
Initialization of the system
Update process of opinion states
Selection of nearest and distant neighbors
Application of external noise (probability q)
Monte Carlo step execution
This structure would allow readers to quickly locate specific details.
2We have followed the above suggestion and modified the text accordingly. We hope the
new description of the algorithm, mostly on page 4 of the revised manuscript, is now clearer
and quickly to locate specific details.
2. Explanation of Variables and Notation
Explicitly define key terms before introducing equations, such as:
μ ij as the affinity between neighbors
q as the probability of external noise affecting interactions
o i as the opinion variable for each site
Including a table of symbols with descriptions could further clarify the notation.
The new second paragraph in section 2.1 has been dedicated to the definition of the
key variables. The new Table 1 has a summary of the corresponding variables and their
respective descriptions.
3. Enhanced Equation Formatting and Context
Equations (1)-(4) should be preceded by a brief intuitive explanation to help readers
understand the transformation process before seeing the mathematical formulation.
This explanation has been done mainly in the second paragraph of section 2.1, where the
updated process was explained in more detail.
Visual representations (diagrams or flowcharts) can illustrate how opinions change step
by step in response to interactions.
Example: A diagram showing a site i interacting with neighbors j and `, demonstrating how
opinion values change.
With this regard we have included in the revised version of the manuscript a new fig. 1,
where the process of updating one site of the network has been detailed described in the
caption of the figure through an example. We hope this extra example can clarify the used
algorithm we have adopted.
4. Conceptual Clarifications
The section states that μ ij is assigned as +1 or -1 with probability q but does not explain
the implication of this choice. Clarifying whether a negative μ ij represents disagreement or
reinforcement of diversity would enhance interpretation.
We have stressed, also in the second paragraph of section 2.1, that negative affinity is in
general disagreement, because it can lead, after updating, to different signs of the opinion
3variable. This is what happens when only ±1 states are possible. In our case there are other
possibilities because of the 0 state present in the opinion variable.
The explanation of why using the workplace lattice alone simplifies computation should
be expanded. Perhaps providing a comparison between the original two-lattice SN model
and this simplified version could highlight computational benefits.
We extended a little the last paragraph of section 2.1 with this regard, including also
some comments from revisor 1.
5. Providing an Example Walkthrough
A short example where an initial set of opinion values is provided, and the updating
rules are applied step-by-step, would make the methodology more digestible.
Example: Starting with an initial configuration of opinions for five sites, applying rules
(1)-(4), and showing the final updated states.
This suggestion turned out to be a nice improvement on understanding the model and
the code. We have thus included a new Figure 1 showing the two steps of updating the
opinion variables of the site in question, its nearest neighbor and a distant neighbor. We
feel better keep in mind a hypercubic lattice instead of a two-dimensional example because
the present work deals with D ≥ 4. We hope such figure could help the same way to better
view the model and methodology.
By implementing these improvements, the methodology section would increase accessibil-
ity for a broader audience, including those unfamiliar with the BCS model while maintaining
scientific rigor.
We hope our modifications are indeed in the direction of improving the accessibility for
a broader audience
Could the Author use the same symbol in figs 2 (b, d ,f) for the derivative dO4/dq as
they have used in figs 2 (a,c,e). Looks that they have used another symbol to represent
order parameter O4 on these figures.
Thanks for drawing our attention to this mistake. It has been corrected in the new figures
2(a)-(c).

Reviewer 4 Report

Comments and Suggestions for Authors

I have carefully read the manuscript entitled Biswas-Chatterjee-Sen model on Solomon networks in (1 ≤ D ≤ 6)-dimensional lattices. I find this work very interesting and I consider it to be very well done. The manuscript is a generalization to 4-6 D of reference 19 where the authors worked with the low dimensionality model 1D to 3D. The work shows consistency with cited works. For a better understanding of the reader, the authors could incorporate a sketch of the design of the networks of dimensions D>3, so that the explanation of the dynamics followed by the system under study is clearer. Given the definition of the model, the phase transition is obtained when q = qc, where the parameter q is equivalent to the temperature for the magnetic systems mentioned. According to the description of the dynamics of the system, the affinity can take values ​​of +1 or -1, this last value with a probability q. Could the authors explain what is the criterion for choosing the values ​​of q or is it done the same as the temperature variation in the magnetic models? The results are clear and consistent with the references cited for the low-dimensional models.

Author Response

I.
REPORT REVISOR 4
I have carefully read the manuscript entitled Biswas-Chatterjee-Sen model on Solomon
networks in (1 ≤ D ≤ 6)-dimensional lattices. I find this work very interesting and I consider
it to be very well done. The manuscript is a generalization to 4-6 D of reference 19 where the
authors worked with the low dimensionality model 1D to 3D. The work shows consistency
with cited works.
We thank the reviewer for a positive evaluation of our paper.
For a better understanding of the reader, the authors could incorporate a sketch of the design
of the networks of dimensions D > 3, so that the explanation of the dynamics followed by
the system under study is clearer.
Since the present work is for D ≥ 4, we could not include a sketch of the network. However,
we indicated, in the beginning of section 2.1, ref. [18] for the reader having an illustration
of SNs in the D = 2 case. In addition, the new Figure 1 can
Given the definition of the model, the phase transition is obtained when q = q c , where the
parameter q is equivalent to the temperature for the magnetic systems mentioned. According
to the description of the dynamics of the system, the affinity can take values of +1 or -1, this
last value with a probability q. Could the authors explain what is the criterion for choosing
the values of q or is it done the same as the temperature variation in the magnetic models?
Yes, it is the same as temperature in magnetic systems. The simulations are done for specific
values of q, with steps ∆q = 0.001 close to the transition.
The results are clear and consistent with the references cited for the low-dimensional models.

Round 2

Reviewer 1 Report

Comments and Suggestions for Authors

The authors have addressed my comments and the manuscript is now improved substantially. 

I just have one doubt still. When a distant neighbor is selected to include the effect of Solomon Network, has it been ensured that for the i-th agent, the same neighbor is selected every time? In a lattice, I would assume that if the i-th agent has coordinate (7,15) (lets assume square lattice), then one of the nearest neighbors should be selected (6,15), (8,15), (7,16) and (7,14) and then another neighbor needs to be selected which is far away. But that far away neighbor must also be selected from 4 such options only, isn't it? Has that been ensured in the algorithm? Either way, I would like that the authors clarify this point in the manuscript.

After taking care of that point, the manuscript can be accepted for publication.

Author Response

Dear Reviewer,   Thank you for your comments   We have revisited our text and from the reviewer's comments we have decided to change Fig. 1 (a new Fig. 1 has been uploaded). With this figure a new description of the model/algorithm has also been made. The new changes are highlighted in blue color in the new revised manuscript.   regards   Welington